# Position: Neural Approximation Is Rarely Justified for Hard Combinatorial Problems

**Pritish Chakraborty** [1]  **Indradyumna Roy** [1 2]  **Soumen Chakrabarti** [1]  **Abir De** [1]

## Abstract

In recent years, there has been a surge in the application of neural approaches to NP-hard combinatorial problems such as subgraph isomorphism, maximum clique and the travelling salesman problem in graphs. These approaches are often evaluated as complete replacements of established combinatorial solver tools, with emphasis on solution quality and runtime. In this position paper, we argue that such wholesale replacements for touted faster inference or better solution quality should not be considered the primary motivation for neural surrogates, and a systematic evaluation of when neural methods are appropriate is required. Given our observations, we contend that in the absence of system-level requirements dictated by the task at hand, such as vector indexing and retrieval, or without the need for end-to-end differentiability, neural surrogates rarely offer compelling advantages over the standard combinatorial solver. In this vein, we develop a comprehensive report of where current neural methods fall short, and subsequently devise a diagnostic checklist for when neural methods are truly applicable.

## 1. Introduction

Many problems of practical interest on graphs and discrete structures are *combinatorially intractable*: the decision space may range over exponentially many *configurations*, such as subsets, permutations, or subgraphs, etc. Examples of these problems span certain archetypal families: clique problems (maximum clique estimation (Marino et al., 2024), subgraph isomorphism (Zeng et al., 2020)), coloring problems (graph coloring (Marx, 2004), max-cut (Hong

et al., 2021)), packing problems (bin packing (Martello & Toth, 1990), job scheduling (Graham, 1966)), path problems such as travelling salesman (TSP) or vehicle routing (VRP) problems (Hoos & Stützle, 2014), and tree optimization problems (Steiner tree search (Robins & Zelikovsky, 2005), and join ordering (Neumann & Radke, 2018)). Many such problems are classified as **NP-hard** (Papadimitriou, 2003). In these problems, it is often the case that no regularities or structural aids exist to cut through the exponential decision space with a polynomial-time algorithm for exact optimal solutions. In practice, however, exact solvers exploit branch-and-bound search, cutting plane methods, and structural decompositions to prune the search space dramatically, solving instances far larger than naive enumeration would permit — though worst-case exponential complexity remains. Further, continuous relaxation of the discrete decision space (as in linear programming or semi-definite programming) is a well-studied and successful tool that reduces the computational difficulty of the problem, while approximating the optimal solution. Examples include semi-definite programming relaxations for max-cut and maximum clique estimation problems (Hong et al., 2021; Lovász, 1979; Grötschel et al., 1984b).

Driven by the success of deep learning methods in the last decade (Krizhevsky et al., 2012; Ronneberger et al., 2015; Gilmer et al., 2017; Vaswani et al., 2017; Devlin et al., 2019), there have been sustained efforts at developing neural approximations for hard combinatorial problems. These neural architectures either aim to replace combinatorial or relaxation based approaches entirely (Vinyals et al., 2015; Bello et al., 2016; Khalil et al., 2017; Kool et al., 2018; Lou et al., 2020), or to augment and enhance existing paradigms (Gasse et al., 2019; Bai et al., 2021), or to preserve end-to-end differentiability of the learning pipeline (Roy et al., 2022b; 2025). Save for the problems where the parameters are unknown and have to be learned (Ferber et al., 2023), most neural approaches are motivated by improving the speed of inference or solution quality.

**In this position paper, we take the contrarian position that neural approaches to solve hard combinatorial problems are rarely justified**. Our position relies on the fact that there are serious challenges holding back the real-world deployment of such neural surrogates, outlined below.

[1]Department of Computer Science and Engineering, Indian Institute of Technology, Bombay, India [2]Department of Computer Science, Aalto University, Finland. Correspondence to: Pritish Chakraborty <pritish@cse.iitb.ac.in>.

*Proceedings of the 43$^{rd}$ International Conference on Machine Learning*, Seoul, South Korea. PMLR 306, 2026. Copyright 2026 by the author(s).

*(a)* Costs considered in our framework.

| Cost type | Description |
|---|---|
| **Per-instance inference cost** | Compute, latency, and memory incurred at test time per instance. |
| **Training cost** | Resources required to train (time, compute, memory). |
| **Data costs** | Effort to obtain labels/solutions (simulator calls, or running exact/approx solvers). |
| **Cost of approximation failure** | Consequences of suboptimality; whether failure modes are tolerable. |
| **Distribution shift cost** | Performance degradation and maintenance overhead as real-world data evolve over time. |

*(b)* Guarantees evaluated in our framework.

| Guarantee | Description |
|---|---|
| **Feasibility** | Output satisfies all constraints and constitutes a valid solution. |
| **Approximation ratio** | Provable lower or upper bounds on approximation ratio. |
| **Robustness** | Stability under perturbations/noise in inputs, parameters, or solver settings. |
| **Incremental improvement** | Predictable scaling with added compute/training; improves reliably or degrades gracefully. |
| **Distributional generalization** | Consistent performance across datasets/domains and under real-world shifts. |

*Table 1.* Summary of the costs and guarantees used to characterize learning-based approaches to combinatorial optimization. We evaluate the merits of various neural approaches for well-known hard optimization problems using this framework.

**Missing framework for costs and guarantees**   Historically, classical algorithm design has responded to the challenge of hard combinatorial problems along a broad spectrum, ranging from purely theoretical to purely computational work. On the more theoretical end, work on approximation algorithms, parameterized complexity and integer programming identifies regimes in which one can obtain worst-case guarantees, either by relaxing the problem, or by exploiting structural properties in the input if present (e.g. bounded tree-width, sparsity). On the other end, practitioners have developed heuristic strategies tailored to specific workloads, and evaluated them in terms of average-case performance over realistic instance distributions rather than the worst-case. Exact solvers such as branch-and-bound and cutting-plane methods exist along the spectrum: they provide both optimality certificates and are practically efficient on real-world instances. In summary, the classical literature spans a rich continuum of approaches, all of which are explicit about the costs they incur and the guarantees they provide. We expand further on classical work in Section 2.

Neural method papers often provide limited accounting of training and label-generation costs, and these are not adequately scrutinized by the community. They either treat these costs as one-time, or simply ignore them. For example, ground-truth label generation alone accounts for having to solve difficult combinatorial problems a large number of times to construct a dataset, imposing a heavy computational cost. Similarly, there is little analysis or discussion of guarantees provided by neural methods, in contrast with the work done by the algorithms community.

To this end, we develop a comprehensive framework based on the imposition of five **costs** and **guarantees**, collectively shown in Table 1. We evaluate several neural methods according to our framework and report which costs were ignored and which guarantees were not provided.

**Role of instance distribution**   Most neural surrogates are introduced with a real-world motivation, but are ultimately trained and evaluated on synthetic instances (or labels). For hard problems, however, the "right" algorithm is inherently instance- and distribution-dependent: a heuristic that performs well on one synthetic regime can fail badly on another. A discriminative neural model can at best learn a heuristic tuned to the training distribution; it is unreasonable to expect it to generalize across all possible instances of an NP-hard problem, including systematically harder or qualitatively different ones. As a result, there is an increased risk of out-of-distribution failures at test time, even when in-distribution performance looks strong, and this risk is rarely acknowledged in how neural surrogates are justified or evaluated.

Conversely, to aid the robustness of the system during deployment, one can introduce out-of-distribution or adversarial examples during training. However, this may increase the training and ground-truth label generation costs several fold, rendering adversarial training infeasible.

**Over-dependence on combinatorial decoders**   Neural surrogates often depend heavily on *combinatorial decoders* that post-process their (typically continuous) outputs into feasible discrete solutions. Many learning pipelines attach non-trivial heuristics or exact routines at the end: local search to repair tours, branch-and-bound to enforce feasibility, maximum matching to enforce consistency, or greedy decoding to construct the final solution. Our analysis of several recent methods reveals that, given reasonable structural or feature-based inputs, they may achieve comparable performance even without a sophisticated neural component. In other words, the decoders do most of the hard work, rendering the eventual solution quality relatively insensitive to

the signals provided by neural models. Yet, empirical evaluations rarely ablate to establish the decoder's contribution properly, and improvements are implicitly credited to the neural surrogate even if the decoder would perform well on its own.

Thus, we make the following contributions.

1. We critically examine existing neural approaches to hard combinatorial problems. Through examples from the literature, and experiments where required, we highlight issues related to consequences of distribution shift and the use of combinatorial output decoders. Additionally, we evaluate these methods based on our costs and guarantees framework.

2. We articulate the limited but important situations in which neural approximations *are* warranted, and clarify what the true end-goal and evaluation criteria should be in these settings (e.g., differentiability, rather than solver replacement).

3. We translate these observations into an actionable checklist that helps practitioners decide when neural surrogates are likely to provide real benefit over classical combinatorial methods.

## 2. Related Work

In this section, we discuss the history of combinatorially hard problems and the efforts to develop various types of approximations for these problems. We then organize the discussion under theory and systems headings for clarity of exposition, but emphasize that these represent ends of a continuum rather than distinct communities. Next, we discuss the relatively recent surge of neural approximators for multiple hard combinatorial problems in three parts: (1) methods that aim to replace the combinatorial solver, (2) methods that interface with or augment the combinatorial solver and (3) methods that prioritize end-to-end differentiability of a larger learning pipeline.

**Hard problems** The concept of NP-hard problems emerged from the seminal early works of Karp and Cook. Cook introduced the notion of NP-completeness and NP-hardness using the Boolean satisfiability problem (Cook, 2023), whereas Karp constructed a catalog of twenty-one diverse NP-complete problems (Karp, 2009), including the travelling salesman and graph coloring problems. These works established that if any one such problem were to be polynomial-time solvable, all of the others would as well through reductions. Garey and Johnson provide a more detailed taxonomy of NP-hard problems across multiple domains (Garey & Johnson, 2002). Over time, the interpretation of NP-hardness has changed to signify problems that are at least as hard as NP-complete ones, including those that are not in NP (Fortnow & Homer, 2003). Researchers have

invested significant effort into understanding what drives the computational difficulty of these problems: inherent combinatorial explosion, several candidate solutions and complex constraint interactions being some reasons.

There exist challenging combinatorial problems that are not NP-hard. These include the assignment problem of finding minimum-cost matching in a bipartite graph (Kuhn, 1955), and the problem of maximum flow in networks (Ahuja et al., 1993). Both are solvable in polynomial time, but are plagued by challenges such as the use of dense and large cost matrices and the need for global structure — for example, maximum matching in general graphs requires Edmonds' blossom algorithm (Edmonds, 1965), and minimum cost flow can be solved via the network simplex method (Ahuja et al., 1993), both of which are considerably more intricate than the graph neural network architectures typically proposed as neural surrogates.

There also exist variants of problems that are not otherwise NP-hard, but become NP-hard due to a slight change in the problem definition. For example, the 2-dimensional assignment problem is polynomial-time solvable, whereas 3-dimensional matching is NP-complete (Dunne, 2008). On a similar note, finding the single source shortest path (Cormen et al., 2022) is not NP-hard, but finding the longest simple path is.

**Classical theory work** A major chunk of work done by the theory community involves the development of approximation algorithms with theoretical guarantees. These algorithms execute in polynomial time and guarantee solutions which are within a known fraction of the optimal solution. Some pertinent examples include the 87.8%-approximation for the max-cut problem using semi-definite programming (Goemans & Williamson, 1995), a 2-approximation algorithm for (weighted) vertex cover via LP relaxation / primal–dual methods (Bar-Yehuda & Even, 1981), the $\mathcal{O}(\log n)$-approximation for the set cover problem (Feige, 1998), and the well-known $1 - 1/e$ greedy approximation for submodular maximization problems (Nemhauser et al., 1978). Further, polynomial-time approximation schemes (PTAS) exist for special cases. This includes Baker's seminal work on NP-hard problems in planar graphs (Baker, 1994), which exploits structural properties in planar graphs (such as bounded treewidth) to obtain close to optimal solutions in polynomial-time. In another example, randomized rounding (Raghavan & Tompson, 1987) is a general technique for converting fractional LP solutions to integral ones and is widely used across covering/packing and network-design formulations.

A complementary point of view is that of *parameterized complexity*. Given input size $n$, one identifies a parameter $l$ (e.g. solution size, treewidth) and aims to develop algorithms exponential in $l$ and polynomial in $n$ (Downey &

Fellows, 2012). A fixed-parameter tractable (FPT) problem is solvable in $f(k) \cdot n^{\mathcal{O}(1)}$, for some function $f$ (Li et al., 2020). These algorithms are exact, but efficient only for smaller parameter values. Finally, one may also utilize structural constraints, such as bounded clique width in graphs, to simplify NP-hard problems. A wide range of hard problems–such as subset sum or partition–are solvable using dynamic programming in pseudopolynomial-time if input data are constrained to be smaller integers (Ibarra & Kim, 1975).

**Classical systems work**   The systems side of the literature emphasizes practicality, with the help of exact solvers that work efficiently not on the worst case, but on typical real-world instances. The operations research community has developed a myriad of tools to solve NP-hard problems in practice. One such method is the branch-and-bound, which systematically explores the solution space by branching on decisions and pruning sub-trees using bounds (Land & Doig, 2009). Modern mixed integer linear programming solvers such as CPLEX and Gurobi incorporate the branch-and-bound and branch-and-cut algorithms. The latter variation utilizes cutting planes, which are valid linear inequalities obtained from constraints, to progressively tighten the given LP relaxation. As an example, the branch-and-cut algorithm features in work on the travelling salesman problem (TSP) (Dantzig et al., 1954). Further, the Concorde TSP solver uses branch-and-cut with sophisticated cut techniques to solve large instances of the TSP (Applegate et al., 2011). Column generation is another similar technique used for large-scale problems (Gilmore & Gomory, 1961).

Local search, heuristics and structural decomposition are some other techniques that are utilized to cut down problem complexity. A heuristic builds a feasible solution often by rules of thumb. An example is the Lin-Kernighan heuristic for the TSP that iteratively performs $K$-optimal edge swaps to improve the length of the tour (Lin & Kernighan, 1973). The method of dynamic programming can solve certain NP-hard problems faster than brute force search by exploiting structure. For example, the Held-Karp dynamic program for TSP and certain scheduling problems runs in $\mathcal{O}(n^2 2^n)$, which is exponential but faster than $\mathcal{O}(n!)$ (Held & Karp, 1962).

**Neural methods for combinatorial problems**   The initial line of work focused on replacing the combinatorial solver with a neural approach. Hopfield and Tank (Hopfield & Tank, 1985) formulated TSP as an energy minimization problem within a Hopfield network, in one of the earliest neural approaches to solving difficult combinatorial problems. A series of such works followed the start of the deep learning revolution, with the work on pointer networks (Vinyals et al., 2015) leading the way. A sequence to sequence model was trained on small TSP instances to produce (empirically) near-optimal TSP tours. Following

this, Bello's work (Bello et al., 2016) trained an RNN-based policy with policy gradients to maximize the total reward of a tour. Their model achieved close-to-optimal results on two-dimensional Euclidean TSP instances of size up to 100 nodes. More such works followed, with the focus on complex routing variants of the TSP (prize-collecting TSP, capacitated vehicle routing, and so on) (Kool et al., 2018) and hard problems on graphs (maximum independent set, coloring) (Khalil et al., 2017).

Among works that seek to augment the solver, learning to guide branch-and-bound is a popular avenue in mixed integer programming. Branching is traditionally a handcrafted heuristic choice in most modern MILP solvers. Gasse et al. (Gasse et al., 2019) trained a graph convolutional network (GCN) to imitate the branching expert's decisions by representing the MILP instance as a bipartite graph with features. Other similar works insert a neural decision-maker inside a local search loop. An example of this is the deep RL framework introduced by Wu et al. (Wu et al., 2021) for routing problems.

Finally, amongst approaches that seek to ensure end-to-end differentiability of a larger learning pipeline, prominent works include OptNet (Amos & Kolter, 2017), which differentiates through the Karush-Kuhn-Tucker conditions of a quadratic program, Gumbel-Sinkhorn networks (Mena et al., 2018), where the Sinkhorn operator enables gradient-based learning for otherwise discrete matching, and Wilder et al's (Wilder et al., 2019a) work on differentiable layers for LPs and integer LPs. Subsequently, Wilder's followup work on differentiable proxies for hard graph problems (Wilder et al., 2019b) and a line of works aimed at graph retrieval (Roy et al., 2022b; Ramachandran et al., 2024; Roy et al., 2022a; Chakraborty et al., 2025; Roy et al., 2025; Jain et al., 2024) demonstrate the use of differentiable neural gadgets for hard combinatorial problems, that can be plugged into a larger learning pipeline.

## 3. Costs and Guarantees

Across much of the surveyed literature, the neural approximator was advertised as either producing high quality solutions, or exhibiting fast inference compared to classical methods. Without a comprehensive discussion of the trade-offs, these claims are incomplete. In this section, we provide a critical treatment of several papers according to the relevant costs and guarantees from the framework discussed in Table 1.

### 3.1. Data costs

A recurring theme across multiple works is the disregard for data costs associated with the neural model. Specifically, papers fail to mention human costs associated with main-

taining representativeness of the train and test data, and the time taken to generate high quality ground-truth labels. For example, the work on pointer networks (Vinyals et al., 2015) trains a sequence-to-sequence model on $n$ uniformly sampled TSP instances. The authors obtain exact ground-truth data by running the Held-Karp algorithm for smaller sized instances, and approximate data by running the Christofides algorithm ($\mathcal{O}(n^3)$) for larger instances. They generate supervision using exact/approximate solvers, and while the paper acknowledges expense, it does not provide a detailed accounting of label-generation cost. The work on efficient GCNs for the TSP (Joshi et al., 2019) uses the Concorde solver (Applegate et al., 2011) to generate ground-truth data for *one million* uniformly sampled training instances. There is a throwaway reference to the infeasibility of generating training pairs for even larger instances, very briefly in the Appendix, in the context of a discussion on supervised learning versus reinforcement learning.

Similar concerns arise for hard graph problems such as subgraph isomorphism (Zeng et al., 2020), maximum clique estimation (Marino et al., 2024), maximum common subgraph (Raymond & Willett, 2002), and graph edit distance (Blumenthal et al., 2020) which have all been the focus of recent neural approximation methods. In a series of works aimed at information retrieval on graphs (Roy et al., 2022b; Ramachandran et al., 2024; Roy et al., 2022a; Chakraborty et al., 2025; Roy et al., 2025; Jain et al., 2024), datasets are constructed by first sampling (corpus and query) subgraphs from real-world datasets (Morris et al., 2020), and then running a suitable combinatorial solver over all query-corpus pairs. The Appendix B has more details on the costs involved in these problems.

### 3.2. Computation costs

While supervised learning methods grapple with label generation costs, methods that use reinforcement learning (RL) face heavy training costs, especially in terms of time taken. For example, Bello et al. (2016) train an RNN using policy gradients over a *million episodes*. While the computational expense is acknowledged, it is claimed as a one-time cost. Nazari et al. (2018) train an encoder-decoder setup using policy gradients with a *large batch size*, but does not quantify or acknowledge the training cost. Structure2Vec (Khalil et al., 2017) proposes a meta-algorithm which uses graph learning and Q-learning (model-free RL) to build solutions. They acknowledge that deep RL architectures are sample inefficient. However, the training cost continues to be large and depends on how many episodes Structure2Vec can train for. Kool et al. (2018) train a transformer network with REINFORCE, and exhibits a similarly high training cost. In this case, there is a *non-negligible inference cost* as well, due to the use of beam search and sampling upto thousands of tours to avoid invalid solutions. POMO (Kwon et al.,

2020) uses an RL approach that exploits symmetry by training on multiple optimal solutions, and the authors claim stability and reduced variance but do not quantify the training cost. Further, unsupervised methods such as the GNN for TSP (Min et al., 2023) also requires a million instances for training.

Most methods account for the per-instance inference cost, but in a number of cases, the dependency on the post-processing algorithm or structure being used is not well quantified. Examples include Joshi et al. (2019), Kool et al. (2018) and Vinyals et al. (2015), where the cost of the beam search post-processing algorithm is not discussed, and POMO (Kwon et al., 2020) which uses several greedy rollouts during inference. Additionally, in most cases, GPU and CPU memory requirements have not been discussed satisfactorily.

### 3.3. Quality guarantees

In Section 2, we discussed multiple classical methods for hard combinatorial problems that explicitly provide theoretical guarantees. In detail, they typically provide approximation guarantees with bounds on the solution, with respect to the optimal exact solution. They also guarantee feasibility by design. In contrast, none of the neural methods discussed so far provide approximation guarantees of any kind. They ensure feasibility by various methods - beam search combined with masking (Vinyals et al., 2015; Kool et al., 2018; Joshi et al., 2019), action masking (Bello et al., 2016; Nazari et al., 2018), by construction (Khalil et al., 2017) or other local heuristics (Ma et al., 2019).

The guarantees provided by these neural approaches are entirely empirical and data-dependent. As a result, their promises are not set in stone, and are subject to change after careful examination. One example of a neural pipeline violating *generalization guarantees* is elaborated by Vinyals et al. (2015). When pointer networks were evaluated on larger (e.g., $n_{\text{train}} = 10$ and $n_{\text{test}} = 50$) or distributionally different instances than those used for training, the networks were unable to generalize. Similar degradation was reported by Joshi et al. (2020), who reported that, even with substantial training and compute, their model, trained on smaller instances, is outperformed by classical heuristics on larger instances. Another example of a violation involves clique number estimation, where the neural pipeline does not offer an *approximation guarantee*. In Karalias et al. (2022), the evaluation protocol measures $f(S)/f(S^*)$, where $S$ is the subset predicted by the model and $S^*$ is the maximum clique subset predicted by the Gurobi solver. However, their neural pipeline does not strictly ensure that $|S| < |S^*|$, with the help of a post processing step or similar. As a result, it is entirely possible to obtain spurious predictions such that $f(S)/f(S^*) > 1$. In contrast to such cavalier treatment in

these works, once an approximation bound is verified in classical theory papers, it is often made tighter by subsequent work.

In contrast to all works surveyed so far, GFNet (Zhang et al., 2023) addresses multiple costs and guarantees suggested above, on problems such as maximum independent set and minimum dominating set on graphs. The authors (1) explicitly report per-instance inference cost and show that GFlowNet is *slower* as it adds one vertex per step, (2) provide a qualitative analysis of GPU memory scaling and limits, (3) guarantee feasibility by engineering the underlying Markov decision process, (4) provide a formal guarantee (under perfect training) that as temperature $T \to 0$, sampling concentrates on optimal solutions, and (5) frame the method as amortizing across graphs via conditional policies and argue this supports generalization to unseen graphs. This establishes GFNet as a promising neural technique for hard combinatorial optimization.

## 4. Data distribution and shifts

While many papers begin with a real-world motivation for their neural approximators, they end up training their models on narrow synthetic distributions. Several early neural TSP/VRP papers spawn the grid (per instance) with which to train their method on by sampling points uniformly at random. The distributional assumptions behind the training data are simplistic, which can lead to methods excelling on in-distribution data but failing to generalize otherwise. For example, Nazari et al. (2018) train their VRP model on synthetic instances where customer locations and demands are sampled at random in the plane, and evaluate it on the same family of random Euclidean instances. There is no evidence that such a model would perform well on more structured real-world distributions, such as clustered urban demand patterns or routes constrained by an underlying road network, because those patterns are never seen during training. The literature has only begun to acknowledge this dissonance: models trained on small or homogeneously distributed instances struggle when presented with larger, realistic instances, or differently structured inputs. Classical operations research (OR) has long used TSPLIB/CVRPLib as standard benchmarks (Reinelt, 1991; Uchoa et al., 2017). Neural methods historically emphasized synthetic Euclidean instances; newer work increasingly reports results on these classical benchmark suites.

**Instance sizes** Further, one common kind of distribution shift is caused by change in instance size. Bogyrbayeva et al. (Bogyrbayeva et al., 2024) point out that for multiple neural TSP papers (Nazari et al., 2018; Bello et al., 2016; Kool et al., 2018; Khalil et al., 2017), different models would have to be trained for different instance sizes, from scratch. These methods incur a *distribution shift cost*, which in turn

leads to further data costs, in that ground-truth data must be regenerated for the given instance size, and the model must be retrained on this data separately. There have been attempts to mitigate this effect in subsequent work. Chen and Tian (Chen & Tian, 2019) introduced NeuRewriter, a local neural rewriting policy, and trained separate models on different VRP distributions (VRP20/Cap30, VRP50/Cap40, VRP100/Cap50). They showed that NeuRewriter outperforms several heuristic and learning baselines on each of these distributions, and that a model trained on one VRP setting can still achieve competitive tour lengths when evaluated on other VRP sizes and capacities. However, in multiple recent TSP/VRP papers (Cheng et al., 2023; Kim et al., 2021; Li et al., 2021), distribution shift issues remain.

**Delicate properties for hard instances** The hardness of a combinatorial problem often hinges on delicate structural properties. For example, in the case of graphs, these could be perfectness, planarity, bounded tree-width, and geometric structure. Mild distribution shifts could dramatically alter the fraction of instances that are truly hard for classical solvers. To our knowledge, existing neural combinatorial optimization works neither characterize nor preserve this easy versus hard aspect of the distribution, which in turn impacts generalization guarantees. The following examples illustrate the range of tractable structural classes that benchmarks may inadvertently sample from.

The TSP is known to be approximated well for the Euclidean space (points in $\mathbb{R}^d$ for fixed $d$) (Arora, 1998). Neural TSP works often sample from these fixed Euclidean spaces. In perfect graphs, the maximum clique estimation and graph coloring problems are solvable in polynomial-time, due to known semi-definite relaxations (Grötschel et al., 1984a). This class of graphs includes bipartite, chordal, interval, comparability and co-graphs (P4-free graphs), among others. Similarly, for graphs with bounded tree-width, a tree decomposition followed by dynamic programming gives a polynomial-time algorithm for maximum clique and maximum independent set (Chakraborty et al., 2020). Further, for certain kinds of intersection graphs, a PTAS is available for maximum clique (Arora, 2003). The subgraph isomorphism problem is polynomial-time solvable on graphs of low tree-width (cactus graphs, interval graphs) or of simpler structure (Eppstein, 2002; Courcelle, 1996), and the vertex cover problem simplifies on many sub-classes of perfect graphs (Brettell et al., 2025). Thus, while these problems are NP-hard in general, they are polynomial-time solvable on a wide variety of instance distributions. Any instance distribution lying in these classes can be solved easily using classical methods. Consequently, without explicitly characterizing the structural properties of the instance distribution, it is impossible to tell whether a neural surrogate is addressing genuinely hard instances or merely operating in regimes where classical algorithms are already efficient.

| Method | Variant | IMDB | RB | Brock | DSJC | MUTAG | Enzymes |
|---|---|---|---|---|---|---|---|
| DIFUSCO | Neural (categorical) | 1.361 | 55.630 | 10.590 | 1.560 | 10.965 | 0.950 |
| | Random | 2.796 | 82.180 | 93.180 | 140.110 | 11.900 | 1.109 |
| | Degree | 2.991 | **47.335** | **6.245** | **0.680** | **8.785** | **0.504** |
| | Core | **0.102** | 69.565 | 53.730 | 65.885 | 10.535 | 0.706 |
| EGN | Neural | 0.102 | 15.615 | 1.310 | **0.030** | **1.010** | 0.109 |
| | Random | **0.000** | **3.270** | **0.560** | 0.035 | 2.715 | **0.008** |
| | Degree | 3.065 | 47.745 | 17.640 | 0.815 | 9.320 | 0.538 |
| | Core | 0.111 | 66.975 | 52.240 | 61.655 | 10.675 | 0.782 |
| SCT | Neural | 4.102 | 50.230 | 35.885 | 92.675 | 11.105 | 0.891 |
| | Random | **0.000** | **14.705** | **1.370** | 2.590 | **3.950** | **0.042** |
| | Degree | 2.185 | 40.955 | 6.605 | **0.740** | 8.740 | 0.269 |
| | Core | 0.074 | 49.935 | 42.240 | 54.160 | 7.190 | 0.311 |

*Table 2.* Mean square error of clique-number estimation for each neural baseline followed by its decoder-only variants (Original) with different input scores (Random, Degree, Core number) on six benchmark datasets. We utilize the decoders shipped with the DIFUSCO (Sun & Yang, 2023), EGN (Erdős goes neural) (Karalias & Loukas, 2020) and SCN (hybrid scattering networks) (Min et al., 2022) methods. For each Method–Dataset combination, the best (lowest) MSE is shown in **bold** and the second-best is underlined. The decoders work *better* given Random and Degree based scores than the neural scores proposed in the respective papers for a majority of cases!

**Other costs and guarantees**  We emphasize that none of these works seriously engage with the *cost of approximation failure*: they report solution quality on benchmark instances, but do not discuss what might happen when the surrogate fails badly in a deployed system. There is typically no robustness guarantee against rare but catastrophic errors, and no analysis of whether suboptimal solutions are merely inconvenient or potentially harmful for the downstream application. While this omission may be benign on toy, synthetic datasets, it becomes critical in real-world settings.

A related omission is that of any *incremental improvement* guarantee. In classical settings, one can often spend additional compute to obtain predictably better solutions, e.g., by increasing search depth, tightening relaxations, or letting a branch-and-bound or local search procedure run longer, with performance that typically improves (or degrades gracefully) as the time budget changes. In contrast, for neural surrogates, it is not always clear whether the output quality may reliably improve with extra test-time compute.

## 5. Decoder dependence

A recurring pattern in neural methods for hard combinatorial problems is the presence of a classical algorithm component, such as a combinatorial decoder or post-processing step at the end. This component ensures feasibility or improves the solution. Empirically, many works do not isolate the contribution of the decoder to the overall performance. For example, Kool et al. (Kool et al., 2018) use beam search with a thousand samples to obtain high quality solutions, showing that the neural network must be supported with a classical algorithm step. In Kim et al. (2021), all methods,

including their own, are evaluated under a fixed ten-second time budget per instance, and performance improves as the number of sampled tours and revision iterations increases. Clearly their best solutions arise from an increasingly powerful decoder layer, not from the one-shot output of the neural model alone. Similarly, POMO (Kwon et al., 2020) performs multiple greedy rollouts during inference, and then selects the best performing solution.

Some earlier work presents preliminary evidence for our decoder dependence argument: Böther et al. (2022) show that for the maximum independent set problem, methods from the literature specifically rely on post-neural tree search procedure to yield high quality solutions, and Santana et al. (2023) show that for the capacitated vehicle routing problem (CVRP), GNN heatmaps did not offer significant advantages over simpler heuristics. We argue that this pattern surfaces much more often in neural combinatorial optimization literature, and claim the general structural dependence of neural surrogates on combinatorial decoders across a broad class of methods.

Given this background, we experiment with the output decoders that accompany various neural methods for the clique number estimation problem. Specifically, we isolate the contribution of the decoder by feeding it simple, non-neural inputs (e.g., structural features or random scores) while keeping the decoding procedure unchanged. We list our findings in Table 2. We note that the decoder alone attains competitive solution quality, often performing better than the original combination of the neural method and the decoder. For example, on the IMDB dataset, the decoders associated with the EGN and SCT neural techniques achieve zero MSE when provided uniform-random inputs. We also experiment with using node features such as degree and

core number as decoder inputs. In the case of DIFUSCO, the degree is the strongest performing heuristic on all but the IMDB dataset. These results suggest that a substantial portion of the empirical gains attributed to "neural solvers" in fact stems from the power of the underlying decoder.

# 6. Discussion and recommendations

The preceding sections painted a critical picture of neural approaches to hard combinatorial problems. Here, we provide guidance on when neural methods are justified, what the primary goals for these methods should be and how practitioners can decide when to use these methods.

**End-to-end differentiability** Our strongest recommendation for neural approximations is to enable end-to-end differentiability within a larger learning pipeline, instead of attempting to replace a combinatorial solver for its own sake. In many applications, the combinatorial problem appears as an *intermediate step*: for example, subgraph isomorphism and maximum common subgraph naturally arise when relevance scores must be obtained for query-corpus graph pairs within a neural graph retrieval system. The downstream objective then is not to solve the combinatorial problem per se, but to retrieve relevant graphs according to task-specific supervision. The combinatorial layer becomes a bottleneck for gradient propagation.

Neural gadgets that reliably approximate the combinatorial problem and support backpropagation are justified even if they are *not strictly faster* than or *not more accurate* than a standalone solver. In the graph search application mentioned above, such backprop may assist fine-tuning text and graph encoders through the retrieval stages.

**System-level constraints** A second justification arises from *system-level constraints* imposed by the downstream task and serving stack—we use "system-level" here to mean engineering and latency requirements, not optimization feasibility constraints. Large-scale retrieval, ranking, and recommendation systems are a prime example. At the modeling level, the underlying allocation problem may involve optimization constraints such as knapsack- and matching-like feasibility requirements (budgets, exposure caps, diversity) — these are handled directly by classical solvers. At the system level, however, online serving can only support a fast "embed-then-ANN" interface: user or query objects are mapped to vectors, corpus items are indexed in an approximate nearest-neighbor structure, and scoring operates via inexpensive and scalable vector operations. In such settings, neural surrogates are not merely competing with an exact combinatorial solver; they are the only practical way to compress a complex utility-maximization problem into embedding functions compatible with the real-time constraints. A similar argument applies to graph retrieval pipelines.

**Neural augmentations within solvers** In this case, the neural method serves as a *heuristic guide* inside a classical solver. Examples include branch-and-bound policies, cut selection and warm-start solutions for local search. These approaches preserve the solver as the ultimate provider of feasibility and approximation guarantees — the neural policy influences the search trajectory but the solver retains optimality certification. This is a sound use of neural methods than outright solver replacement. Nevertheless, the remaining costs in our framework still apply–we provide some examples below.

On data and training costs: Khalil et al. (2016) collect expert demonstrations via Strong Branching, which is expensive to compute per node; Gasse et al. (2019) require solving large families of MILP instances for supervision — one-time costs that should be reported explicitly. On distribution shift: Khalil et al. (2016) train on a fixed family of MILP instances, meaning the learned policy may not transfer to structurally different problem families; Gasse et al. (2019) generalize within a problem family but not across structurally different ones — a gap highlighted by cross-domain evaluations in Huang et al. (2024), which show that ML-guided branching policies trained on single distributions perform poorly on held-out domains. On incremental improvement: a neural branching policy alters the search trajectory in ways that may not degrade gracefully outside the training distribution, even if optimal convergence is theoretically preserved. In summary, neural augmentations within solvers represent the most defensible use case among solver-replacement alternatives, but practitioners should still account for supervision costs and distributional assumptions before deployment.

Based on these points, we provide a **checklist of questions** to answer when deciding whether a neural approximation is required. (1) Is a discrete (hard) combinatorial layer posing a bottleneck within a neural pipeline? (2) Do system-level constraints impose the need for neural embedding representations? (3) Can the combinatorial solver itself be augmented with neural capabilities? (4) If a neural replacement is attempted, are the various costs accounted for, and are any guarantees provided? (5) Have simple decoder and heuristic style solutions been attempted as strong baselines?

# 7. Alternative Views

A principled opposing perspective to our position is that of *amortized optimization* (Amos, 2023). From this standpoint, the training and data costs that we highlight in Section 3 are infrastructure investments: any neural surrogate must pay once to learn the input distribution, and then amortize that cost over potentially hundreds of thousands of inference-time calls. Thus the correct unit of comparison is not the cost of a single classical solver call against a neural inference call, but the total cost of the neural surrogate over its deployment

lifetime. If the instance distribution is stable, and large-scale repeated inference is central to the application–as in industrial recommendation systems and logistics planners– the amortized cost per query of a neural surrogate may be substantially cheaper than repeatedly invoking a classical solver.

Our critique is that the fixed-distribution assumption is a fragile one. The relevant deployment distribution is often unknown in advance (Bengio et al., 2021), and the distribution shift forces retraining. Furthermore, as we discuss in Section 4, neural surrogates may degrade non-gracefully under shift, making the assumption of a stable distribution particularly consequential. Therefore, the amortization viewpoint is defensible only when the practitioner can maintain that the training and deployment distributions will remain close during deployment lifetime, and when the volume of inference-time calls is large enough to justify the upfront investment. Our checklist in Section 6 is designed precisely to make these conditions explicit, so that the amortized argument can be evaluated rather than assumed.

A second opposing viewpoint presents when the optimization problem itself is not fully specified. Often the mapping from observed data to a well-defined objective is *unknown*: the data are partially observable, the generative process is latent, and what we observe are *decisions* produced by an implicit mechanism. In such regimes, learning is necessary to identify the model itself. This is the area of inverse optimization: from near-optimal decisions, infer the underlying cost parameters that rationalize them (Ahuja & Orlin, 2001). One example is when we observe discrete matchings or permutations but cannot access the costs that shaped them. While Optimal Transport (Peyré et al., 2019) formalizes a forward map from costs to optimal couplings, the inverse-OT problem learns costs from observed couplings, making learning necessary even when the forward problem is tractable (Stuart & Wolfram, 2020). Similarly, SurCo (Ferber et al., 2023) argues for learning surrogate costs when the true downstream loss is nonlinear or only indirectly observed, training a linear surrogate objective end-to-end against the true task loss. SurCo demonstrates strong performance on three real-world tasks where the objective is genuinely unknown: GPU embedding table sharding, inverse photonic design, and non-linear route planning with stochastic travel times. From this perspective, neural methods are justified precisely because the objective must be learned from data — the role of the neural component is cost inference, not solver replacement.

## 8. Conclusion

To conclude, we argue that end-to-end neural "solver replacements" for NP-hard combinatorial problems are too often evaluated as if faster inference or marginally better benchmark numbers were sufficient justification. In practice, what matters is the *full tradeoff*: the cost of generating labels and training at scale, the reliance on powerful combinatorial decoders that may be doing most of the work, the lack of guarantees beyond empirical performance, and the brittleness that appears under shifts in instance size or structure. Our costs-and-guarantees framework is meant to make these tradeoffs explicit and comparable across methods, and our decoder analyses illustrate why ablations and budget-matched evaluations are essential before crediting improvements to the neural component. Ultimately, neural approaches are most compelling when they address a genuine missing ingredient, such as enabling differentiability in a larger pipeline, or operating under constraints where classical solvers cannot.

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

## A. Declaration of LLM Usage

We confirm that the content of this paper is written entirely by us, with LLMs used only for polishing and repeated iterations of edits.

## B. Costs and Combinatorial Solver Details for Graph Problems

Here, we discuss the specifics of the graph retrieval line of works mentioned in Section 2. The works on subgraph matching and inverted indexing for graphs (Roy et al., 2022b; Chakraborty et al., 2025) use the VF2 family of algorithms (Cordella et al., 2004) to generate ground-truth data for subgraph matching. At its core, the VF2 family of algorithms uses a depth-first backtracking search to find optimal solutions. Similarly, the work on maximum common subgraph estimation (Roy et al., 2022a; Bai et al., 2020) uses the Glasgow solver (McCreesh et al., 2020), which generates MCS ground-truth data by branch-and-bound algorithms, and the graph edit distance works (Jain et al., 2024; Ranjan et al., 2022) use GEDlib (Blumenthal et al., 2019), which provides a variety of algorithms, including branch-and-bound and integer programming. In the worst case, these algorithms are exponential-time, and fit the mold of the classic systems style mentioned in Section 2. Additionally, in the more recent works out of these, the training set consists of anywhere from a hundred thousand to a million labeled query-corpus graph pairs. Clearly, to generate ground-truth labels, the underlying hard combinatorial problem must be solved several times using classical solvers, raising concerns about the impact of unacknowledged costs.

## C. Decoder Experiment Details

In this section, we expand on the experimental details for the decoder ablation experiments in Section 5.

### C.1. Datasets

We experiment on the following real-world datasets (Sanokowski et al., 2023; Morris et al., 2020): IMDB-BINARY (IMDB), Enzymes, and Mutagenicity (Mutag). Additionally, we also experiment on these synthetic datasets (Johnson & Trick, 1996; Xu et al., 2007): DSJC, Brockington and RB. In the case of Mutag, the clique number represents the maximum common induced subgraph's size. In DSJC, the graphs take a k-partite form with at least one k-clique. In Brockington, each graph is constrained to contain cliques with nodes of low connection or degree. Finally, the RB dataset is modelled on randomized hard constraint satisfaction with difficult to locate cliques.

### C.2. Setup

We evaluate the decoders that support the neural methods in DIFUSCO (Sun & Yang, 2023), the "Erdős goes neural" paper (Karalias & Loukas, 2020) and the hybrid scattering networks paper (Min et al., 2022). Specifically, we first report the original MSE results for each of these methods from (Roy et al., 2025), where the neural method is supported by its corresponding decoder, having reproduced these results in our environment. Next, we replace the neural outputs with the following heuristic outputs: scores sampled uniformly at random from $\mathcal{U}(0, 1)$, the degrees of the nodes and the core numbers of the nodes. We obtain the degrees and core numbers using a mix of utilities available in torch-geometric (Fey & Lenssen, 2019), and self-written code.

These heuristic outputs are fed to the corresponding decoder, and the MSE is computed for the clique numbers obtained against the ground-truth. In the case of the random heuristic, we sample eight times and take the best performing clique result out of these for each test set graph.

### C.3. Methods

**DIFUSCO** This paper introduces continuous and discrete diffusion modelling for hard combinatorial optimization problems. The authors originally solve the maximum independent set problem, and we adapt their solution for the clique number estimation problem. This method is the only on our list to use the maximum clique itself as supervision. The output decoder that follows the neural pipeline is a standard greedy clique decoder. The original MSE results that we report are for the discrete diffusion model (Sun & Yang, 2023).

**Erdős goes neural (EGN)** Inspired by the Erdős probabilistic method, the authors train a graph neural network to output scores on subsets of nodes. They claim that their loss optimization framework guarantees good integral solutions for the

given constraints. Their output decoder draws from randomized rounding techniques and deterministically decodes a feasible solution from the given neural outputs (Karalias & Loukas, 2020).

**Hybrid scattering networks (SCT)** The authors train a graph neural network with a multi-component loss function that encourages node subsets that are connected and form cliques. The probability output vectors for the node subset are passed into a rule-based decoder for the final, feasible prediction (Min et al., 2022).

## C.4. Hardware

We perform all our experiments on a single RTX A6000 GPU, on an Intel(R) Xeon(R) Gold 6130 CPU @ 2.10GHz server running Debian 12 with 64 CPU cores and 1.6TB physical RAM.

## D. Additional Experimental Details From Rebuttal Discussion

In this section, we discuss experimental results presented to reviewers during the rebuttal period.

### D.1. Classical scoring vs. neural scoring in graph retrieval

*Table 3.* Full-corpus oracle scoring vs. LSH+Oracle on PTC datasets, showing retrieval speedup with preserved MAP@100.

| Dataset | Oracle time/q (s) | LSH+Oracle time/q (s) | LSH+Oracle MAP@100 |
|---------|-------------------|-----------------------|---------------------|
| PTC-FR | 26.07 | 0.915 | 1.00 |
| PTC-FM | 49.34 | 1.153 | 0.98 |
| PTC-MR | 26.75 | 1.146 | 1.00 |

We illustrate the role of system-level constraints with an example from graph retrieval. The retrieval system is tasked with returning the $K$ most relevant corpus graphs in sub-linear time (Chakraborty et al., 2025). This requires an index built over the corpus using neural embeddings, so that a small but high quality candidate set may be retrieved. To quantify the benefit of neural embeddings at the indexing stage, we compare full-corpus oracle scoring against LSH+Oracle scoring. A FAISS-based random hyperplane LSH index built on IsoNet neural embeddings (Roy et al., 2022b) is used to first filter the corpus to at most 2500 candidates, and then these candidates are scored.

Table 3 summarizes the results. LSH+Oracle achieves a 20–40× speedup over full-corpus oracle scoring while preserving a near-perfect mean average precision of 1. Without neural embeddings (and subsequently, the index), full-corpus oracle scoring is too slow for deployment. This confirms our checklist item number 2: neural approximation is justified due to the system-level constraints imposed by the (retrieval) task.

*Table 4.* LSH+Oracle vs. LSH+Neural scoring on PTC datasets (100K corpus, 100 test queries, candidate subset ≤ 2500).

| Dataset | LSH+Oracle time/q (s) | LSH+Oracle MAP@100 | LSH+Neural time/q (s) | LSH+Neural MAP@100 |
|---------|------------------------|---------------------|------------------------|---------------------|
| PTC-FR | 0.915 | 1.00 | 0.149 | 0.62 |
| PTC-FM | 1.153 | 0.98 | 0.152 | 0.50 |
| PTC-MR | 1.146 | 1.00 | 0.174 | 0.63 |

However, the justification for neural approximation does not extend to the scoring stage. We replace the oracle scorer with IsoNet and compare MAP@100 and per-query scoring time on the same LSH-filtered candidate sets. Table 4 shows that the neural scorer reduces per-query time by about 6× relative to the oracle, but at a steep quality cost: MAP@100 drops by 0.35–0.5 across all three datasets. The oracle already operates under one second per query on the filtered candidate set, making it the better choice.

### D.2. Hidden training costs of neural methods in the clique number problem

Neural surrogates for hard combinatorial problems are often defended on the basis that training is a one-time cost, to be amortized over several forward passes. However, this argument is valid only when deployment-time data distribution closely matches the training-time distribution. If a distribution shift is present, the benefit of amortization is negated, as the training

*Table 5.* Per-instance inference time vs. total training time for EGN and DIFUSCO-Cat across graph datasets (trained on RTX A6000 with early stopping).

| Dataset | EGN infer/q (s) | EGN train | DIFUSCO-Cat infer/q (s) | DIFUSCO-Cat train |
|---------|-----------------|-----------|-------------------------|-------------------|
| IMDB | 0.117 | ~26 h | 0.789 | ~1 h |
| DSJC | 0.152 | ~19 h | 0.686 | ~1 h |
| Brock | 0.151 | ~9 h | 0.796 | ~3.5 h |
| Enzymes | 0.087 | ~2.5 h | 0.789 | ~4 h |
| RB | 0.201 | ~15.5 h | 0.906 | ~3.5 h |
| MUTAG-m | 0.473 | ~23 h | 0.827 | ~14.5 h |

cost must be borne repeatedly for each new data regime. Additionally, the absolute magnitude of training costs is rarely reported alongside inference costs in neural combinatorial optimization literature.

In Table 5, present these costs–per-instance inference time alongside total training time–for two representative clique number estimation methods (Roy et al., 2025), EGN and DIFUSCO, trained on six graph datasets with an early stopping limit of 10K epochs on a single RTX A6000 GPU. Training times vary from 1 hour to 26 hours even with early stopping, while per-instance inference times are sub-second across all datasets.

### D.3. Neural degradation under distribution shift

*Table 6.* Clique number estimation MSE for EGN under default and OOD (out-of-distribution by instance size) splits on PTC-MM-m, compared against graph-structural baselines.

| Dataset | EGN | Random | Degree | Core | ClusterCoef |
|---------|-----|--------|--------|------|-------------|
| PTC-MM-m (Default) | 0.284 | 0.745 | 2.396 | 1.943 | 0.859 |
| PTC-MM-m (OOD) | 3.260 | 1.359 | 4.255 | 3.513 | 1.160 |

*Table 7.* Clique number estimation MSE for SCT under default and OOD splits on PTC-MM-m, compared against graph-structural baselines.

| Dataset | SCT | Random | Degree | Core | ClusterCoef |
|---------|-----|--------|--------|------|-------------|
| PTC-MM-m (Default) | 1.802 | 0.402 | 1.838 | 1.072 | 0.556 |
| PTC-MM-m (OOD) | 1.256 | 0.818 | 3.406 | 1.707 | 0.636 |

In Tables 6 and 7, we provide experimental evidence of distributional fragility using clique number estimation on the PTC-MM dataset (Roy et al., 2025). We first construct an out-of-distribution (OOD) split by partitioning instances according to size, rendering the test distribution different from the training distribution. Then we report MSE between predicted and true clique numbers for EGN and SCT, alongside four non-neural baselines: random assignment, degree, core number and clustering coefficient.

For EGN, the neural MSE increases more than tenfold under OOD shift ($0.284 \rightarrow 3.260$), while the random baseline degrades more gracefully ($0.745 \rightarrow 1.359$). SCT, on the other hand, is already outperformed by random and clustering coefficient baselines on the default split. Additionally, it exhibits a decrease in MSE under OOD shift ($1.802 \rightarrow 1.256$). This non-monotonic behaviour serves to further emphasize the opacity of neural generalization. Unlike classical solvers which provide certificates of optimality and well-known failure modes, neural surrogates can behave unpredictably.

# E. Classical vs. Neural Solvers: A Comparative Analysis

We provide a systematic comparison of classical solvers and neural surrogates across the cost and guarantee dimensions introduced in Table 1. This comparison makes explicit why classical solvers impose zero training costs and negligible data costs, while neural surrogates inherit and amplify both.

*Table 8.* Comparison of classical solvers and neural surrogates across the cost and guarantee dimensions of our framework.

| Dimension | Classical solvers | Neural surrogates |
|---|---|---|
| Per-instance inference cost | Controllable via time limits, node limits, or gap tolerances (Land & Doig, 2009). | Inference is fast but post-processing (e.g., beam search, greedy rollouts) adds cost that is not always quantified (Section 3 (3.2)). |
| Training cost | Not applicable. | Ranges from moderate (supervised) to very high (RL); rarely reported in full (Section 3 (3.2)). |
| Data cost | Cost of processing data into a form accepted by the solver is generally low. | Requires running classical solvers thousands to millions of times to generate labels (Section 3 (3.1)). |
| Cost of approximation failure | Bounded and transparent: duality gaps or optimality certificates available even under early termination. | Opaque: no internal signal indicates when a prediction is unreliable (Geisler et al., 2021). |
| Distribution shift cost | Instance-agnostic; no retraining required when the problem regime changes. | High: retraining and label regeneration required for new instance sizes or structures (Joshi et al., 2020; Bogyrbayeva et al., 2024). |
| Feasibility | Guaranteed by construction: branch-and-bound, cutting-plane, and LP-relaxation (Land & Doig, 2009). | Not guaranteed by the neural component; enforced by post-hoc decoders or masking heuristics (Section 5). |
| Approximation ratio | Often provable: e.g., 0.878 for max-cut (Goemans & Williamson, 1995), 2 for vertex cover (Bar-Yehuda & Even, 1981). | Quality often empirical and data-dependent (Section 3 (3.3)). |
| Robustness | Failure cases identifiable through gap analysis; behavior under perturbation is well-studied. | Sensitive to small perturbations; adversarial fragility documented for SAT and TSP (Böther et al., 2022). |
| Incremental improvement | Predictable: additional compute yields monotonically tighter bounds in branch-and-bound (Land & Doig, 2009); local search produces non-worsening tours (Lin & Kernighan, 1973; Helsgaun, 2000). | Unclear: RL training suffers from sparse rewards and high variance (Bello et al., 2016; Kool et al., 2018). |
| Distributional generalization | Not applicable: no training distribution to generalize from. | Often poor: models trained on one regime fail on another (Section 4). |

It is striking that classical solvers impose zero training costs and negligible data costs while neural surrogates inherit and amplify both. The "one-time cost" argument put forward in defence of neural methods holds only if the deployment-time distribution closely matches the training-time distribution — and such closeness is not guaranteed in general. Joshi et al. (2020) show that neural TSP solvers trained on small instances are unable to generalize to larger instances at practical scales, and Bogyrbayeva et al. (2024) document that for multiple neural TSP methods, different models must be trained for different

instance sizes from scratch, re-incurring training and data costs each time. More broadly, the relevant distribution of problem instances for a given real-world task is often not known in advance (Bengio et al., 2021), making the fixed-distribution assumption fragile.

A further advantage of classical solvers is the transparency of failure. Branch-and-bound solvers progressively tighten a pair of primal and dual bounds throughout their search, and the duality gap certifies solution quality at any point during execution (Land & Doig, 2009) — including under early termination. Neural surrogates offer no analogous signal: there is no internal indicator of when a prediction is unreliable, making deployment-time failure silent and difficult to detect.

