# OpenReview forum: "Position: Neural Approximation Is Rarely Justified for Hard Combinatorial Problems"
_ICML.cc/2026/Position_Paper_Track — ICML 2026 Position Paper Track regular_

### Official Review · Reviewer_6dsy · 2026-02-25

**Significance:** 3
**Argument Clarity:** 3
**Rating:** 5
**Confidence:** 3

**Questions:**

Please reply to my comment listed in "Weakness".

**Alternative Views Section:**

Yes

**Compliance With Llm Reviewing Policy A Conservative:**

Affirmed.

**Discussion Potential:**

3

**Final Justification:**

The authors have addressed my comments. Since my score is already positive, I plan to keep it as is.

**Paper Summary:**

NP-hard combinatorial optimization problems such as the traveling salesman problem cannot be solved by polynomial-time algorithms. In recent years, many neural algorithms have been developed specifically to address the speed and quality of solutions. As a position paper, the authors take a stance on the application of neural network methods in hard combinatorial optimization problems, arguing that they are generally not applicable in solving hard combinatorial optimization problems and do not offer significant advantages over traditional solvers. The authors identify the following challenges for neural network methods: lack of cost and guarantee analysis mechanisms, the influence of specific instance distributions, and over-reliance on combinatorial decoders. To address these challenges, the authors propose a cost and guarantee framework to help researchers better analyze whether neural methods have objective benefits.

**Position:**

Yes

**Position In Title:**

Yes

**Related Work:**

3

**Strengths And Weaknesses:**

## Strengths
+ The authors pointed out three challenges faced by neural methods for solving hard combinatorial optimization problems: the lack of cost and guarantee analysis mechanisms, reliance on specific instance distributions, and excessive dependence on decoders.

+ The authors have provided a detailed framework for costs and guarantees to assist researchers in conducting a more comprehensive analysis of neural methods. Based on this framework, the authors have analyzed the existing neural methods for solving hard combinatorial optimization problems from multiple perspectives.

+ The authors conducted some experimental analyses, and the results showed that most of the benefits of many neural methods actually originated from the decoder. This indicates that the existing neural methods have an excessive reliance on the performance of the decoder.

+ The authors present a series of questions to assist researchers in better analyzing whether it is worthwhile to employ a neural approximation method.


## Weakness
- In the analysis section of the cost and guarantee framework, although the author has comprehensively analyzed various neural methods, it seems that no comparison with traditional algorithms has been provided in this regard. It is suggested to include some comparative results, which will be more conducive to supporting the author's position.

**Support:**

3

---

> ### Author Rebuttal · Authors · 2026-03-31
>
> We thank the reviewer for their insightful comment and plan to update the paper accordingly. We provide a sketch in the form of a comparison table below.
>
> | Dimension | Classical solvers | Neural surrogates |
> |---|---|---|
> | Per-instance inference cost | Controllable via time limits, node limits, or gap tolerances [A]. | Inference is fast but post-processing (e.g., beam search, greedy rollouts) adds cost that is not always quantified (ref. Section 3.2) |
> | Training cost | Not applicable. | Ranges from moderate (supervised) to very high (RL); rarely reported in full (ref. Section 3.2) |
> | Data cost | Generally low. | Requires running classical solvers thousands to millions of times to generate labels (ref. Section 3.1) |
> | Cost of approx. failure | Bounded and transparent: duality gaps or optimality certificates even under early termination [B][A]. | Opaque: no internal signal indicates when a prediction is unreliable [C]. |
> | Distribution shift cost | Instance-agnostic; no retraining required when the problem regime changes. | High: retraining and label regeneration required for new instance sizes or structures [D][E]. |
> | Feasibility | Guaranteed by construction: branch-and-bound, cutting-plane, LP-relaxation [B]. | Not guaranteed by the neural component; enforced by post-hoc decoders or masking (ref. Section 5). |
> | Approximation ratio | Often provable: e.g., 0.878 for max-cut [F], 2 for vertex cover [G]. | Quality often empirical and data-dependent (Section 3.3). |
> | Robustness | Failure cases identifiable through gap analysis; behavior under perturbation studied [H]. | Sensitive to small perturbations; adversarial fragility documented for SAT and TSP [C]. |
> | Incremental improvement | Predictable: additional compute yields monotonically tighter bounds in B&B [A]; local search produces non-worsening tours [I][J]. | Unclear: RL training suffers from sparse rewards and high variance [K][L]. |
> | Distributional generalization | Not applicable. | Often poor: models trained on one regime fail on another (ref. Section 4). |
>
> Classical solvers impose zero training and negligible data costs while neural surrogates inherit and amplify both. The "one-time cost" argument holds only if deployment and training distributions are close — which is not guaranteed. [D] and [E] document this failure for neural TSP solvers across instance sizes. More broadly, the relevant distribution for a real-world task is often unknown in advance [M], making the fixed-distribution assumption fragile. We also note that transparency of failure is a key advantage of classical solvers: branch-and-bound maintains a primal-dual bound pair throughout search, so the duality gap certifies solution quality at any point, including under early termination. Neural surrogates offer no analogous signal.
>
> We demonstrate the data and inference cost tradeoff concretely in graph retrieval. We use three PTC datasets from [N], each with 100K corpus graphs and 100 test query graphs. A random-hyperplane LSH index is constructed using the FAISS library with ISONet [O] embeddings, and for each query a subset of at most 2500 corpus graphs is retrieved. Both the VF2-based oracle (NetworkX) and ISONet are then tasked with scoring this filtered subset. We report MAP@100 and per-query scoring time:
>
> | Dataset | LSH+Oracle time/q | LSH+Oracle MAP@100 | LSH+Neural time/q | LSH+Neural MAP@100 |
> |---------|------:|------:|------:|------:|
> | PTC-FR |0.915 s|1.00|0.149 s|0.62|
> | PTC-FM |1.153 s|0.98|0.152 s|0.50|
> | PTC-MR |1.146 s|1.00|0.174 s|0.63|
>
> The oracle achieves ~1 MAP at the cost of under a second to ~1 second more per query. The neural model is faster at inference time but MAP drops sharply on all datasets — by 0.35–0.50. This raises the question of whether the neural model offers any benefit at all, given that (1) it requires the classical solver to generate training labels, and (2) the inference time difference is modest even under these favourable conditions. Finally, we note that training ISONet on these PTC datasets took time ranging from 3 days to 7 days. Training was done on a single RTX A6000 GPU with early stopping on a max of 10K epochs. These training times are hidden costs that should be scrutinized.
>
> [A] Land & Doig, 2009.
>
> [B] Conforti et al., 2014.
>
> [C] Geisler et al., 2022.
>
> [D] Joshi et al., 2020.
>
> [E] Bogyrbayeva et al., 2024.
>
> [F] Goemans & Williamson, 1995.
>
> [G] Bar-Yehuda & Even, 1981.
>
> [H] Lu et al., 2023.
>
> [I] Lin & Kernighan, 1973.
>
> [J] Helsgaun, 2000.
>
> [K] Bello et al., 2016.
>
> [L] Kool et al., 2018.
>
> [M] Bengio et al., 2021.
>
> [N] Chakraborty et al., 2025.
>
> [O] Roy et al., 2022.

---

> > ### Author Rebuttal · Reviewer_6dsy · 2026-04-01
> >
> > Thanks for the responses! Since my rating is already positive, I decide to remain my rating unchanged.

---

### Official Review · Reviewer_qydK · 2026-03-09

**Significance:** 4
**Argument Clarity:** 4
**Rating:** 5
**Confidence:** 4

**Questions:**

- **(Q1):**
> For hard problems, however, the “right” algorithm is inherently instance- and distribution-dependent: a heuristic that performs well on one synthetic regime can fail badly on another. A discriminative neural model can at best learn a heuristic tuned to the training distribution;

Don't both the neural and the classical solver have the same issue here? A neural network trained on one distribution (a heuristic for a specific data distribution) will struggle on a completely different distribution.

- **(Q2):** Are you aware of any real-world issues where neural approximators have been used in a justified setting (mainly for ill-specified problems as in Section7) and achieved strong performance?


### Mischellaneous
- (M1) One paper by Roy does not have a year in the bibliography.
- (M2) "Next, we discuss the relatively recent surge of neural approximators for multiple hard combinatorial problems in three parts **-** (1) (...) ". I would suggest ":" instead of "-" here.
- (M3) "Given input size $n$, one identifies a parameter $l$ (solution size, treewidth)". I would suggest "e.g. solution size or treewidth".
- (M4) "In Karalias et al. (Karaliaset al., 2022), (...)". I would suggest "In \citet{ ...}"

**Alternative Views Section:**

Yes

**Compliance With Llm Reviewing Policy A Conservative:**

Affirmed.

**Discussion Potential:**

4

**Final Justification:**

The rebuttal did adequately address my concerns, thus I am in favor of accepting this paper.

**Paper Summary:**

This paper argues that current neural approximation is rarely justified: (1) it requires significant training costs and vasts amount of data that needs to be generated by using classical solvers thousands to millions of time. (2) They provide no quality guarantees and are not guaranteed to provide better results if run / trained for longer. (3) They are very dependent on data distribution and vulnerable to data shifts. Finally, (4) the authors find that in many current approaches a classical algorithm is used as a decoder. They show experimentally, that most of the performance gains is due to this classical algorithm and not the neural network.

**Position:**

Yes

**Position In Title:**

Yes

**Related Work:**

3

**Strengths And Weaknesses:**

**(S1 Importance & Discussion Potential):** If neural approaches are only rarely able to outperform classical approaches, we should be aware of this and devote less research effort on them.

**(S2 Framework)** The proposed framework with which neural approximation methods should be analyze is compelling. In particular:
- Data costs: the compute costs of creating training data should be considered and reported. In particular, if creating the training data requires one million examples generated by solving instances  with a classical solver, this would mean that replacing the neural approximator by a classical solver would basically make the first one million inference calls free (from a runtime perspective).
- Data distribution / shifts: generating good synthetic data is difficult and existing datasets could benefit from a rigorous analysis of their structural properties as small changes in the structure can make problems significantly easier to solve. This means, it is often not clear whether a neural solver can solve hard instances or is just evaluated on easy instances.

**(S3 Experimental Evidence):** The experiment on decoder dependence is extremely interesting. Neural methods often still use a classical algorithm component such as  combinatorial decoder. The authors demonstrate, that this decoder often is the source of the strong performance and not the neural component. For this, instead of outputs of a neural network they input different features (such as random noise) into the decoder and show that this often significantly outperforms the case of using neural outputs.

**(S4 Discussion Potential - When They are Justified):** The idea that neural approximators are mainly justified in settings where the problem is not fully specified (Section 7) is interesting and leads to a very complete narrative: The authors determine that in most cases neural approximators are not justified (due to training costs, weaknesses based on data, or lack of guarantees) but also describe a compelling scenario where they are justified.

**(W1 Experimental Evidence):** This paper could be strengthened with experimental evidence that supports its main point: the hidden cost of training neural approximators (Section 3) and the impact of data distribution/shift (Section 4). While there is some experimental evidence in this paper) it is about a slightly tangential issue (see S3). Thus, while Sections 3 and 4 give a good overview of the literature, they could have significantly more impact if there were some experiments to back up the claims.

**(W2 Alternate Views):** The alternate views section could be extended. It argues that there exist situations where the problem is not well specified and thus that classical approaches cannot be used. However, this section could be strengthened by going into more directions _or_ considering more related papers (currently there are 4 total citations in this section see Q2). Some other possible directions: computational cost of generating data (section 3.1)  or training (section 3.2) is only a one time cost.

**Overall,** this is a good paper with a clear narrative and position. The (small) experiments support the idea that neural approximators are indeed "rarely justified". While the paper could benefit from more experimental evidence and a better alternate views section this is still a good paper: weak accept. If the authors address my concerns I will increase my score significantly.

**Support:**

3

---

> ### Author Rebuttal · Authors · 2026-03-31
>
> We thank the reviewer for their pertinent comments and address each of them below.
>
> > (W1 Experimental Evidence): This paper could be strengthened with experimental evidence...hidden cost...more experiments.
>
> **Hidden training costs.** We contrast previously unreported training times against inference times for two clique number estimation methods (trained on RTX A6000, with early stopping active):
>
> | Dataset | EGN (infer / train) | DIFUSCO-Cat (infer / train) |
> | ------- | ------------------- | --------------------------- |
> | IMDB | 0.117 s / ~26 h | 0.789 s / ~1 h |
> | DSJC | 0.152 s / ~19 h | 0.686 s / ~1 h |
> | Brock | 0.151 s / ~9 h | 0.796 s / ~3.5 h |
> | Enzymes | 0.087 s / ~2.5 h | 0.789 s / ~4 h |
> | RB | 0.201 s / ~15.5 h | 0.906 s / ~3.5 h |
> | MUTAG-m | 0.473 s / ~23 h | 0.827 s / ~14.5 h |
>
> Neural practitioners often assert that the training time is a one-time cost, but this assertion holds only if the deployment-time data distribution closely matches the train-time distribution. If the assumption does not hold, then the training cost will have to be borne repeatedly.
>
> **Distribution shift.** Three independent studies illustrate the brittleness of neural methods: (1) [A] (Fig. 3) show that neural TSP solvers trained on TSP-20 fail to generalize to larger sizes, and models trained on TSP-100 generalize poorly to smaller sizes. (2) [B] (Fig. 3) show that for max-cut, training on the toroidal distribution and testing on others (e.g., weighted ER) causes the approximation ratio to drop below 25%, versus near 1.0 in-distribution. (3) [C] show that on subgraph edit distance, GREED-25's RMSE on CiteSeer queries of size 25–50 is 9.052 vs. 0.948 for the in-distribution GREED-50 model — a ~10x degradation.
>
> **New experiments on distribution shift.** We experiment with clique number estimation on PTC-MM using an OOD split (by instance size) from [D]. We report MSE between predicted and true clique numbers:
>
> | Dataset | EGN | Random | Degree | Core | ClusterCoeff | PageRank |
> | ------------------ | --------- | ------- | ------ | ----- | ------------ | -------- |
> | PTC-MM-m (Default) | **0.284** | *0.745* | 2.396 | 1.943 | 0.859 | 2.615 |
> | PTC-MM-m (OOD) | 3.260 | *1.359* | 4.255 | 3.513 | **1.160** | 4.318 |
>
> | Dataset | SCT | Random | Degree | Core | ClusterCoeff | PageRank |
> | ------------------ | ----- | --------- | ------ | ----- | ------------ | -------- |
> | PTC-MM-m (Default) | 1.802 | **0.402** | 1.838 | 1.072 | *0.556* | 2.074 |
> | PTC-MM-m (OOD) | 1.256 | *0.818* | 3.406 | 1.707 | **0.636** | 3.495 |
>
> For SCT, the neural method is outperformed on both splits by random and clustering coefficient inputs. There is an unexpected MSE decrease for the neural method. For EGN, the neural MSE degrades more steeply than random under shift. These results highlight the lack of generalization bounds for neural methods on hard combinatorial problems.
>
> > (W2 Alternate Views): The alternate views section could be extended...
>
> We plan to revamp this section. A coherent counter-view is that of *amortized optimization* [E]: training and data costs are infrastructure investments that amortize over millions of future instances, and the right comparison is the full pipeline, not a single instance. This becomes strongest when the instance distribution is stable and large-scale repeated inference is central. It does not negate our critique, but is a principled opposing perspective we will develop.
>
> > (Q1): For hard problems...
>
> Classical solvers are not expected to generalize across distributions, and the community characterizes this explicitly. In contrast, the neural CO literature rarely tests for distributional limitations; training and testing on the same synthetic distribution creates an impression of general-purpose applicability. Multiple studies confirm this asymmetry: [F] observe that neural heuristics "trained on instances with certain characteristics underperform when tested on instances with different characteristics," and [G] show that neural solvers for SAT and TSP are fragile under small adversarial perturbations, noting that standard evaluation protocols are "too optimistic." We believe this asymmetry needs to be called out.
>
> > (Q2): ...real-world issues...justified setting...
>
> [H] learn linear surrogate costs for ill-specified combinatorial problems (SurCo), evaluating on three real-world tasks: (1) embedding table sharding for GPU assignment, (2) inverse photonic design, where the objective requires expensive nonlinear electromagnetic simulations, and (3) non-linear route planning with stochastic travel times. SurCo outperformed domain expert baselines and generic non-linear solvers across these tasks.
>
> > Miscellaneous
>
> We will address these inconsistencies during the camera-ready revision.
>
> [A] Joshi et al., 2020.
>
> [B] Nath & Kuhnle, 2024.
>
> [C] Ranjan et al., 2022.
>
> [D] Jain et al., 2024.
>
> [E] Amos, 2022.
>
> [F] Manchanda et al., 2022.
>
> [G] Geisler, Sommer et al., ICLR 2022.
>
> [H] Ferber et al., 2023.

---

> > ### Author Rebuttal · Reviewer_qydK · 2026-04-01
> >
> > Thank you for your reply and the additional experiments. Assuming the authors do indeed plan to revamp the alternate views section (see also reviewer 1uKV) I am now in favor of accepting this paper (weak accept $\Rightarrow$ accept).

---

### Official Review · Reviewer_a2tK · 2026-03-13

**Significance:** 3
**Argument Clarity:** 4
**Rating:** 5
**Confidence:** 3

**Questions:**

1. What are specific use cases or domain the authors expects neural solvers are useful based on the proposed criteria?
2. Neural augmentation within solvers is a promising idea (some research has been done on this for MIP solvers), how does this paradigm fits into your proposed costs-and-guarantees framework?

**Alternative Views Section:**

Yes

**Compliance With Llm Reviewing Policy A Conservative:**

Affirmed.

**Discussion Potential:**

4

**Paper Summary:**

This paper argue that end-to-end neural solver replacements for NP-hard combinatorial problems are often evaluated only by their inference performance without taking account the full tradeoff: generating labels using conventional solvers, lacks of guarantees beyond empirical performance, and lacks of generalizability. The proposed costs-and-guarantees framework aims to make the neural-conventional tradeoffs explicit and comparable across methods.

**Position:**

Yes

**Position In Title:**

Yes

**Related Work:**

4

**Strengths And Weaknesses:**

# Strength

- The paper addressed an important issue that has been in the community's discussion for several years: The trade-offs between using conventional solvers versus collecting data and train a neural solver. Furthermore, issues such as the data distribution and problem size are discussed in detail.
- The paper proposed a comprehensive framework to address the raised issue.
- The paper provided a comprehensive context by giving sufficient citations.

# Weakness

- Lacks experimental data to support their position. The position is stated on a wide range of problems, yet experimental results are small and uninteresting.

**Support:**

2

---

> ### Author Rebuttal · Authors · 2026-03-31
>
> We thank the reviewer for the constructive feedback.
>
> > Lacks experimental data
>
> (1) **Classical vs. neural scoring in graph retrieval**. To emphasize the impact of system-level constraints in graph retrieval, we construct a FAISS-based random-hyperplane LSH on three PTC datasets from [A] (100K corpus graphs, 100 test query graphs, LSH-filtered subset size ≤2500), and compare VF2 (NetworkX oracle) against ISONET [B] for scoring the filtered subset. The LSH index is constructed using embeddings obtained from ISONET.
>
> |Dataset|LSH+Oracle time/q|LSH+Oracle MAP@100|LSH+Neural time/q|LSH+Neural MAP@100|
> |-|-:|-:|-:|-:|
> |PTC-FR|0.915 s|1.00|0.149 s|0.62|
> |PTC-FM|1.153 s|0.98|0.152 s|0.50|
> |PTC-MR|1.146 s|1.00|0.174 s|0.63|
>
> However, we observe that the MAP drops sharply for the neural model — by 0.35–0.50 across datasets — while the oracle recovers ~1 MAP with under a second to ~1 second of additional scoring time. This shows that, once system-level constraints are satisfied, the purported benefit of fast neural inference is dampened significantly by quality degradation, consistent with our argument in Section 3 that per-instance inference cost must be evaluated jointly with solution quality.
>
> **(2) Distribution shift.** We evaluate EGN and SCT on in- and out-of-distribution variants of PTC-MM [C], reporting MSE for clique number estimation. The OOD variant contains graphs drawn from a structurally different (by size) distribution than the training set:
>
> | Dataset | EGN | Random | Degree | Core | ClusterCoeff |
> |---------|--:|--:|--:|--:|--:|
> | Default | **0.284** | *0.745* |2.396|1.943| 0.859 |
> | OOD | 3.260 | *1.359* | 4.255 | 3.513 | **1.160** |
>
> | Dataset | SCT | Random | Degree | Core | ClusterCoeff |
> |---------|--:|--:|--:|--:|--:|
> | Default | 1.802 | **0.402** | 1.838 | 1.072 | *0.556* |
> | OOD | 1.256 | *0.818* | 3.406 | 1.707 | **0.636** |
>
> EGN's MSE rises sharply OOD (0.284 → 3.260) while the random baseline degrades far more gracefully (0.745 → 1.359). SCT fails to outperform simple heuristics on either variant. The unexpected MSE drop for SCT under OOD (1.802 → 1.256) further illustrates the unpredictable nature of neural generalization — the network does not degrade monotonically, making its failure modes hard to anticipate or bound, which is precisely the distributional generalization concern raised in Table 1 of our framework.
>
> > Specific use cases for neural solvers
>
> Neural methods are warranted when system-level constraints make classical solvers infeasible at the indexing stage (checklist item 2). In large-scale graph retrieval, the system must return the K most relevant corpus graphs in sub-linear time. This requires indexing N corpus items via embedding representations so that approximate nearest-neighbor search can produce a small candidate set efficiently. To produce high-quality embeddings for subgraph-isomorphism-based retrieval, one must approximately solve an NP-hard problem — which is where a neural surrogate is justified. The neural model is not used at the final scoring stage; the classical solver handles scoring once the candidate set is small enough. To quantify this benefit, we compare full-corpus oracle scoring against LSH+Oracle using ISONET [B] embeddings for filtering:
>
> | Dataset | Oracle time/q | LSH+Oracle time/q | LSH+Oracle MAP@100 |
> |---------|------:|------:|------:|
> | PTC-FR |26.07 s|0.915 s|1.00|
> | PTC-FM |49.34 s|1.153 s|0.98|
> | PTC-MR |26.75 s|1.146 s|1.00|
>
> The LSH+Oracle achieves a 20–40× speedup while preserving ~1 MAP. Without neural embeddings, full-corpus oracle scoring is too slow for real-time deployment. Our checklist confirms that neural approximation is justified as it is the only practical way to compress the retrieval problem into a form the solver can handle under real-time constraints.
>
> > Solver augmentation and costs/guarantees
>
> The key structural advantage is that the solver provides **feasibility and approximation guarantees**. Both [D] and [E] operate inside branch-and-bound — the neural policy selects branching variables while the solver retains optimality certification, unlike replacements such as beam search or post-hoc repair. Our framework's remaining costs still apply. **Data/training costs**: [D] collect expert demonstrations via Strong Branching (expensive per node); [E] require solving large MILP families for supervision — one-time costs that should be reported explicitly. **Distribution shift**: [D] train instance-specifically, mitigating shift at per-instance overhead cost; [E] generalize within a problem family but not across structurally different ones [F]. **Incremental improvement**: a neural branching policy alters the search trajectory in ways that may not degrade gracefully outside the training distribution, even if optimal convergence is theoretically preserved.
>
> [A] Chakraborty et al., 2025
>
> [B] Roy et al., 2022
>
> [C] Jain et al., 2024
>
> [D] Khalil et al., 2016
>
> [E] Gasse et al., 2019
>
> [F] Huang et al., 2024

---

> > ### Author Rebuttal · Reviewer_a2tK · 2026-04-06
> >
> > I would like to thank the authors for their detailed rebuttal. The additional experiments, especially LSH+Oracle and LSH+Neural retrieval comparison are interesting results. I acknowledge these additions and maintain my previous score for the submission.

---

### Official Review · Reviewer_1uKV · 2026-03-16

**Significance:** 3
**Argument Clarity:** 1
**Rating:** 2
**Confidence:** 4

**Questions:**

Please address the issues that I have described above in your rebuttal.

**Alternative Views Section:**

No

**Compliance With Llm Reviewing Policy A Conservative:**

Affirmed.

**Discussion Potential:**

3

**Final Justification:**

One of my main concerns with acceptance is the matter of novelty, given that two papers have presented exactly the same point previously. I believe that this warrants a discussion among the reviewers, for which reason I am keeping my score.

**Paper Summary:**

This paper argues that using neural approaches for solving combinatorial optimization problems is not as useful as one would expect. The authors emphasize that neural approximators must often be followed by decoders for producing feasible solutions. However, the work done by such decoders can be more significant to the quality of the result than what is accomplished by using a neural encoder in the first place, as illustrated with some experiments on existing frameworks in Section 5. The authors also argue about papers in this line of work not accounting for the costs of obtaining data and training models, the potential impact of distribution shift by merely changing the size of the problem being solved, and the fact that the family of instances used in benchmarks may not reflect the difficulty of solving the problem in general.

**Position:**

Yes

**Position In Title:**

Yes

**Related Work:**

1

**Strengths And Weaknesses:**

## Strengths

1. The authors paint a sober and, from my point of view, mostly accurate perspective on neural approximators: they can be beneficial, but we can only tell that with proper reporting and sufficient ablations. Depending on who you ask, their position is not as contrarian as they claim.

2. The authors present a very extensive literature review. Going back to 1985 with neural approximations was great. At the same time, there are some chunks of the optimization literature review which, even if correct, do not seem central to their argument.

3. The authors make a valid case for using neural approximation for the heuristics within solvers, since this intervention may speed up the end-to-end process while not voiding optimality guarantees of MILP solvers.

## Weaknesses

1. Although interesting, the results reported in Section 5 are not new. For example, consider this ICLR 2022 paper [https://arxiv.org/abs/2201.10494] and this CPAIOR 2023 paper [https://arxiv.org/abs/2210.12075]. The first show how random inputs to a decoder can lead to similar results, and the second shows that simpler OR heuristics are as good as neural approximators. Perhaps these results are less known than they should, and that is certainly a reason for pushing for more discussion, as the authors intend to.

2. The characterization of the optimization area and its community needs refinement:
   - Dividing the optimization community between theory and systems is very reductive. First, this is not a terminology used in optimization. Similarly, I have never seen "mixed integer linear optimization **pipelines**" used in any optimization paper. Second, there is plenty of theory surrounding branch-and-bound, cutting planes, and even heuristic algorithms under the right conditions. Hence, it would be more accurate to consider member of this community as in a spectrum from purely theoretical to purely computational work. These are not entirely separate communities.
   - The way that this divide is presented also misses the point that valid bounds and certificates of optimality are part of the artifacts produced by the algorithms of this so-called systems community within optimization. This is worsened by statements such as "[t]his leaves exhaustive enumeration as the only choice for exact optimal solutions, in the worst case". While the qualifier after the comma makes it true, in practice this is not what happens. The size of problems solved to optimality by solvers would be impractical if mere enumeration was the case.
   - Likewise, it is a wild guess to claim that the major chunk of work done in theory is with approximation algorithms.
   - The idea that some combinatorial problems are challenging even if not NP-hard just because they are in some sense more difficult to implement is a little strange - especially if we consider that the same could be argued about the sophisticated methods used for neural approximation, such as GCNs.

3. I disagree with the authors about neural approximators being helpful when constraints need to be enforced. A neural approximator alone cannot enforce constraints, but an MILP formulation can. Even if proving optimality is a lengthy process, there is plenty of general-purpose heuristics - such as the feasibility pump and derived works - enabling MILP solvers to quickly find good quality solutions, which makes it possible to use them under strict time limits. Moreover, if a decoder is going to be needed, why going to the trouble of using a neural approximator if ultimately the decoder will need to implement an heuristic anyway? While I know that disagreeing with a position is not grounds for rejecting the paper, I believe that making a claim about what works and what doesn't without proper benchmarking risks taking future work in the wrong direction.

4. The Alternative Views section covers exceptions to the framing proposed by the authors, not an alternative perspective that directly disagree with their claim. Hence, I cannot accept this section as what is required for a position paper.

## Other comments

1. A lot of what is discussed under "classical theory work" and "classical systems work" in Section 2 is not directly relevant to the goal of this position paper. Likewise, a big chunk of the second paragraph under "delicate properties for hard instances" in Section 4 is not directly relevant: from "In perfect graphs," to "(Brettell et al, 2025).", the entire presentation is about special cases of optimization problems without any claim that these particular cases reflect a cherry-picking in neural approximation papers, like claimed in the first example of that paragraph. With that said, the first two sentences and the remainder of this paragraph are great.

2. The role attributed to inverse optimization in Section 7 could also be filled by papers self-describing their topic as constraint learning or optimization over trained neural networks, not to mention the potential role of contextual optimization here.

3. Many uses of stand-alone hyphens in the text should have been em-dashes (--- in LaTeX).

## Summary

Although I agree with the position of the authors in general terms and I see significant strengths in their argument, I believe that this paper needs further refinement. I believe that the authors should start by laying out their checklist in the first page of the paper, develop their argument around validating that checking, make a clear stance of how someone may oppose what they are claiming, and portray the optimization community in a more accurate way. Without a clear description of how optimization problems are solved in the absence of neural networks, including the conditions under which optimality and feasibility can be assured (which are more generous than currently portrayed in the paper), the authors would not do justice on building an intuition in their readers about when neural approximation is warranted or not.

**Support:**

3

---

> ### Author Rebuttal · Authors · 2026-03-31
>
> We thank the reviewer for their detailed feedback. Our rebuttal is concise due to the character limit, but we will surely provide an elaboration of our arguments in the revised paper (esp. the alternate views section).
>
> > Dividing .. theory and systems
>
> The division was unintentional. We will reword to clarify that existing work ranges in a continuum between theory and practice, and that neural approaches straddle a range within this continuum. “Pipeline” has become colloquial, often used for systems that are not pipelines, but is easily removed.
>
> > valid bounds...optimality...artifacts
>
> We are aware of some of the theoretical guarantees around branch-and-bound, cutting planes, and heuristics [C, D]. Some guarantee optimal solutions within finite (but not polynomial in the worst case, unless the problem size is bounded [E, F]) time. Others are instances of polynomial-time approximation algorithms, with an early example being the celebrated 3/2-Approximation for Metric TSP [G, H]. While the body of work is immense and diverse, polynomial time approximation has been the focus in recent two decades. We agree with the reviewer that “exhaustive enumeration” paints an unfairly grim picture, and will reword this. We also recognize that improved bounds can speed up convergence or improve approximation quality. One reasonable alternative view (subject to experimental validation) is that effective neural networks can provide superior bounding methods within a combinatorial framework. Our experience with recent systems that “decode” neural representations into a solution do not match this template.
>
> > some problems are challenging even if not NP-hard, just because they are in some sense more difficult to implement
>
> We intend to discuss some of the branch-and-bound, cutting plane, or other delicate problem-specific strategies (such as approximating group Steiner trees [A], facility location, or metric labeling [B]), which are algorithmically more difficult than GCN. We understand the wording should be improved.
>
>
> > decoder…reported in Section 5 are not new. ICLR 2022, CPAIOR 2023
>
> Thanks for pointing out these papers, which certainly deserve more visibility. They additionally strengthen our claim that decoders do the heavy lifting rather than neural networks. We will include and discuss them.
>
> We observed that decoders that are much simpler than tree search routines contribute significantly by themselves and therefore, we believe such a discussion would benefit the community.
>
> > what is discussed … in Section 2 is not relevant to the goal of this position paper.
>
> Thanks for calling this out; it seems the line of argument was lost during the writing process. Edited version:
> Neural solvers are broadly understood to exploit average-case complexity over data distributions that are relevant in practice. How do these distributions relate to the structural properties (or lack thereof) that render individual instances tractable or intractable? We enumerated several well-known easy and hard structural characteristics for illustration. A further question is whether the empirical performance of neural solvers can be attributed in any straightforward way to the composition of such structures within the training and evaluation data. Another concern is that sampling or generating semi-synthetic data is particularly precarious in the presence of such structures.
> > System-level constraints L401–L412
>
> Thanks for pointing out that “constraints” has been used in multiple senses (system and optimization). One system constraint example is that we have an item scoring and ranking problem to solve, not a decision problem. Another constraint may be that the number of items to be ranked is too large for exhaustive corpus scan, requiring dense indexes (Section 6) supported by neural representation learning. Optimization or formulation constraints may arise from a need for ranking diversity, fair exposure in recommender systems, or marketing budgets, which are handled directly by optimization APIs and often via objective tweaks in learning systems.
>
> > Alternative view missing
>
> In Section 6, we highlighted cases where neural solvers may be justified. We will refrain from repeating these in the alternative view section. Instead, one counterargument would be as follows:
>
> Neural solvers are useful because costs are multifarious and not easily aggregated. While a combinatorial solver starts afresh with each problem instance (and then may spend more time), a neural solver pays upfront (in terms of labeled data and computation) by learning to exploit narrow non-worst-case instance distributions, which is then exploited during faster inference. Depending on the level of domain shift (not) required, neural solvers may form the optimal strategy in some circumstances.
>
> [A] Garg et al., 2000
>
> [B] Kleinberg & Tardos, 1999
>
> [C] Land & Doig, 1960
>
> [D] Gomory, 2009
>
> [E] Lenstra, 1983
>
> [F] Hochbaum and Shmoys, 1987
>
> [G] Christofides, 1976
>
> [H] Serdyukov, 1978

---

> > ### Author Rebuttal · Reviewer_1uKV · 2026-04-02
> >
> > I appreciate the effort of the authors in addressing most of my feedback in the little space that they had (I don't understand why the ICML rebuttal needs to be so constrained).
> >
> > Although not central to the paper's argument, it is upsetting that the authors insist on conflating combinatorial optimization theory with approximation algorithms. There are books that barely touch on approximation algorithms, such as Integer Programming by Michele Conforti, Gérard Cornuéjols, and Giacomo Zambelli (https://link.springer.com/book/10.1007/978-3-319-11008-0). Second, take a look at the list of accepted papers at IPCO 2026 (https://events.math.unipd.it/ipco2026/accepted-papers): there are 4 papers on approximation algorithms, a little over 10% of the total, in contrast to 6 about matroids alone.
> >
> > With that said, one of my main concerns with acceptance is the matter of novelty, given that two papers have presented exactly the same point previously. I believe that this warrants a discussion among the reviewers, for which reason I am keeping my score.

---

### Decision · Program_Chairs · 2026-04-30

**Decision:**

Accept (regular)

**Comment:**

All reviewers are positive about the clear presentation of the multiple practical challenges in deploying neural approximations of combinatorial solvers, and the high-quality literature review.  While experiments are not the focus of this submission, some are included and the strengthen the support of their points.  One more negative review has concerns about two uncited papers which also voiced (more specific and narrow) concerns about the performance of neural solvers.  Overall however, these papers do not seem to be sufficient reason to reject this submission; please do cite them and discuss in your final manuscript.